# Prototype or Exemplar Representations in the 5/5 Category Learning Task

**DOI:** 10.3390/bs14060470

**Published:** 2024-05-31

**Authors:** Fang Chen, Peijuan Li, Hao Chen, Carol A. Seger, Zhiya Liu

**Affiliations:** 1Center for Studies of Psychological Application, School of Psychology, South China Normal University, Guangzhou 510631, China; 2019010204@m.scnu.edu.cn (F.C.); 2021010235@m.scnu.edu.cn (P.L.); h0uch4n@gmail.com (H.C.); 2Department of Psychology, College of Education and Sports Sciences, Yangtze University, Jingzhou 434023, China; 3Department of Psychology, Molecular, Cellular and Integrative Neurosciences Program, Colorado State University, Fort Collins, CO 80523, USA

**Keywords:** category learning, 5/5 category structure, prototype representation, exemplar representation, computational model

## Abstract

Theories of category learning have typically focused on how the underlying category structure affects the category representations acquired by learners. However, there is limited research as to how other factors affect what representations are learned and utilized and how representations might change across the time course of learning. We used a novel “5/5” categorization task developed from the well-studied 5/4 task with the addition of one more stimulus to clarify an ambiguity in the 5/4 prototypes. We used multiple methods including computational modeling to identify whether participants categorized on the basis of exemplar or prototype representations. We found that, overall, for the stimuli we used (schematic robot-like stimuli), learning was best characterized by the use of prototypes. Most importantly, we found that relative use of prototype and exemplar strategies changed across learning, with use of exemplar representations decreasing and prototype representations increasing across blocks.

## 1. Introduction

Categorization is a fundamental human ability that enables us to comprehend the underlying organization of our surroundings, facilitating suitable responses. Extensive research has been conducted on various category structures, encompassing categories defined by single-factor rules, multiple discrete-valued attributes, and/or continuously varying attributes. Our specific focus lies on multidimensional categories characterized by discrete features, which resemble the task structures commonly observed in numerous categorization studies, such as the classic Shepard [1] tasks, family resemblance tasks [2], and the 5/4 task [3]. In all these tasks, the stimuli involved possess numerous discrete features that can be utilized for categorization.

Prototype and exemplar theories are the prevailing explanations for how individuals acquire knowledge about discrete multidimensional categories. In line with prototype theories [4,5,6], categories are represented by their central tendency, known as the prototype, which is derived from specific exemplars and encompasses all defining features. When confronted with a new instance, individuals assign it to the category whose prototype bears the closest resemblance. On the other hand, exemplar theories [3,7,8,9,10] propose that category learning relies on memorized representations of individual exemplars. When encountering a new sample, individuals categorize it based on its similarity to previously encountered exemplars belonging to different categories.

In our study, we used a new discrete multidimensional category learning task, the 5/5 task, which is a variant of the highly studied 5/4 task [3,5,11,12,13,14,15]. We used multiple methods to assess for each participant whether performance was characterized by exemplar or prototype representations, including responses on critical diagnostic stimuli, and computational modeling. We also examined how representation use changed across the time course of learning.

### 1.1. The 5/4 and 5/5 Category Learning Tasks

One of the most commonly used discrete multidimensional category learning tasks is the 5/4 task [3]. In this task, participants learn two categories, A and B, through trial and error. The categories vary along four binary-valued dimensions, as illustrated in Table 1. Only nine total stimuli, five from A and four from B, are included in the training set to allow for tight control of prototype and exemplar similarity.

One way the 5/4 task is used to distinguish between prototype and exemplar representations is through examination of responses to two diagnostic stimuli, A1 and A2, which prototype and exemplar theories categorize differently. According to prototype theory, the prototype for Category A is A0(1111), because 1 is the most common value on each dimension in Category A, and the prototype for Category B is B0(0000), because 0 is the most common value on each dimension in Category B. Thus, because A1 (1110) shares three features with the A prototype but only one with the B prototype, the prototype model predicts that A1 will be classified as a member of Category A more often than A2 (1010) will, which shares two features with both prototypes. In other words, the prototype theory predicts an A1 advantage over A2 in classification performance. Exemplar theory, in contrast, predicts an A2 advantage, because whereas A2 shares three features with two Category A members (A1 and A3) and two or fewer with any Category B member, A1 shares three features with only one other Category A member (A2) and two Category B members (B1 and B2). The use of A1 vs. A2 categorization to identify category learning strategy is supported by a large number of previous studies [13,14].

However, the 5/4 category structure includes an ambiguity for Category B in dimension 2: across the four B stimuli, the 0 and 1 values occur equally often. Most research assumes that the prototype for Category B is B0 (0000), but it could equally likely be (0100) based solely on the B Category stimuli. This difference does not affect the validity of the use of A1 or A2 to identify strategy, but it does result in a very weak diagnostic weight for dimension 2. Computational modeling studies using the 5/4 task showed that participants’ attention weight for dimension 2 was negligible (for example, the weight was 0.049 in Liu et al. [16] on a scale ranging from 0.0 to 1.0). We eliminated this ambiguity by adding a fifth stimulus, exemplar B5(1001) in Category B, as shown in Table 1, and refer to this version as the 5/5 task. The addition of B5 eliminates the ambiguity about the prototype of Category B as B0(0000), and increases the diagnostic value of dimension 2, which we hypothesized would make the prototype structure more salient and result in the use of prototype strategies for learning.

### 1.2. Factors Influencing Reliance on Prototype and Exemplar Representations

Prior research has primarily focused on identifying evidence supporting either prototype or exemplar strategies, assuming that a single category learning function underlies all categorization tasks. However, other findings suggest the existence of multiple category learning strategies that can lead to distinct representations with varying qualities [17,18,19]. Within the area of discrete multidimensional category learning, researchers have presented evidence that performance can be fit by both prototype and exemplar models [20], and that different neural regions may underlay learning of each [21,22]).

A complete account of the factors determining when and whether prototype or exemplar representations are formed during learning of discrete multidimensional categories has not yet been established, but there are some intriguing hints. Most prominently, people have examined different types of category structures. Bowman and Zeithamova [23] manipulated category coherence (closeness of stimuli in the training set to the prototype) and found that more coherent categories were more likely to lead to prototype representations and less coherent categories to exemplar representations. In the 5/5 category structure, we argue that the addition of the fifth stimulus would serve to increase the coherence of the category and, thus, participants would be likely to learn the 5/5 category structure using prototype strategies.

### 1.3. Computational Modeling

Computational modeling has brought many exciting results to category learning research [24,25,26,27]. We adapted two well-established computational models to identify category representations: the generalized context model (GCM, [28,29]) and the multiplicative prototype model (MPM, [4,5]). The models were also employed to determine the dimensions that participants prioritized the most in their decision-making processes and to compare these weighting preferences across different strategies.

### 1.4. Current Study

In the current study, our hypothesis was that the 5/5 category structure would result in participants relying on prototype representations given the increased coherence of the category structure present in the 5/5 task. Thus, we predicted that, in the experiment, participants would tend to learn prototypes, and those who used prototypes would be more likely to be successful at learning. We further examined how representations changed across the time course of learning by fitting the prototype and exemplar models separately across blocks of trials. We hypothesized that use of the prototype strategy would become more dominant as learning progressed.

## 2. Methods

### 2.1. Participants

Participants included 44 undergraduate students, 23 females and 21 males, aged between 18 and 23. None had previously participated in related category learning experiments. All had normal vision or corrected to normal vision. Participants were paid for their participation. According to G*Power calculations, the experiment required at least 30 participants to reach the medium effect size (1−β = 0.8; *α* = 0.05). Therefore, the number of participants in this experiment reached the standard.

### 2.2. Design

We examined category learning in the 5/5 task for the schematic “robot”, illustrated in Figure 1. In the robot, each feature is distinct and instantiated by a simple geometric form (e.g., triangles or rectangles for ears). Although features are physically connected, they do not form parts of a familiar figure. The images are black and white line drawings without texture, shading, or color.

### 2.3. Materials

Stimuli were formed using the 5/5 category structure illustrated in Table 1 and Figure 1. Stimuli consisted of line drawings of robots that differed along four possible binary features: antennae (straight or curly), ears (triangular or rectangular), eyes (triangular or circular), and base (white square or black oval). A total of 10 different stimuli were formed from each prototype following the feature values in Table 1. Dimensions were randomly assigned to features and counterbalanced across participants.

### 2.4. Procedure

The procedure was similar to previous studies using a 5/4 category structure in that participants viewed individual stimuli, and learned to categorize them via trial and error with feedback [3,5,11,12,13,14,15]. In each trial, one of the stimuli from the set of 10 possible stimuli was randomly presented in the central position of the computer display screen. The screen was set to a resolution of 1024 × 768 and stimuli size of 6 × 8 cm. Participants were asked to categorize the stimulus by pressing the F (for Category A) or J (for Category B) keys on the keyboard. Each stimulus was presented until the participant made a response, at which time the stimulus was replaced by informative feedback (the words “Right” or “Wrong” in Chinese, 对 and 错, respectively). Maximum stimulus presentation time was 5 s; if a participant did not respond within this time, the trial was terminated and no response was collected. Feedback was displayed for 1 s followed by a fixation cross during a 500 ms intertrial interval. The experiment was divided into blocks of 10 trials, within which each stimulus was presented once in random order. Participants continued performing the task until they reached a criterion of accuracy at or above 90% on three consecutive blocks, or completed a total of 70 blocks without reaching criterion.

### 2.5. Computational Modelling

We applied the two computational models described in the Introduction, the MPM and GCM to participants’ data.

#### 2.5.1. The GCM

The exemplar-based GCM represents categories by storing exemplars in memory. It classifies items by first calculating the weighted distance between to-be-classified item i and an exemplar stored in memory, j, by
(1)$$d_{ij}=C[\sum_{k=1}^{N}w_{k}\left|x_{ik}-x_{jk}\right|]$$
where $x_{ik}$ takes the value 0 or 1 depending on dimension k. Differences among dimensions are weighed by a dimensional decision weight (attention weight), $w_{k}$ (0 ≤ $w_{k}$ ≤ 1, Σ$w_{k}$ = 1). C is an overall sensitivity parameter that determines the rate at which similarity declines with distance (see [3,8], for more extensive discussion). C was initialized to a range between 0 and 20 to meet the requirements of the genetic algorithm modeling function and was subsequently optimized during the modeling process. Distance is transformed by the following exponential decay function:(2)$$\eta_{ij}=e^{{-d}_{ij}}$$

Equation (2) produces a similarity between exemplars i and j [1]. Similarity between an exemplar from Category A is $\sum_{j\in A}\eta_{ij}$, and an exemplar from Category B is $\sum_{j\in B}\eta_{ij}$.
(3)$$P_{\left\langle R_{A}|S_{i}\right\rangle }=\frac{\sum_{j\in A}\eta_{ij}}{\sum_{j\in A}\eta_{ij}+\sum_{j\in B}\eta_{ij}}$$
which states that the probability of a Category A response R given item i is the ratio of the similarities between that item and all stored exemplars j in Category A versus those in Categories A and B. Similarities are raised to a gamma parameter specifying the level of deterministic responding. The value of the Gamma parameter was determined by reference to previous research. In [29], p. 926 and [13], p. 829, the value is set to 1. A gamma of 1 does not change the result of the equation so it is not included in Equation (3) (mathematically, any value raised to a power of 1 is unchanged). In the GCM, deterministic responding is represented by the gamma parameter. Although a gamma parameter may also be added to the MPM in the following equation, Equation (6), one cannot estimate its value independently of the sensitivity parameter. In fact, versions of the MPM with and without the gamma parameter are mathematically identical (see [29], p. 926, for a complete proof).

#### 2.5.2. The MPM

The MPM is similar to the GCM in that items are classified via distances in a multidimensional space. However, distances in this model are calculated between to-be-classified items and a prototype rather than with previously classified exemplars.
(4)$$d_{ipJ}=C[\sum_{k=1}^{N}w_{k}\left|x_{ik}-P_{kJ}\right|]$$
where $P_{kJ}$ is the value of the category J prototype on stimulus dimension k. Otherwise, the distance calculation functions much like Equation (1). As with the GCM’s Equation (2), distance is transformed into similarity by the exponential decay function by
(5)$$\eta_{ipJ}=e^{{-d}_{ipJ}}$$
where *C* represents the participant’s sensitivity to similarity [29]. Similarity to the Category A and B prototype is then entered into the choice equation given by
(6)$$P_{\left\langle R_{A}|S_{i}\right\rangle }=\frac{\eta_{ipA}}{\eta_{ipA}+\eta_{ipB}}$$

We used a Genetic Algorithm to fit the data, with 3000 iterations for parameter estimation, and sum of square deviation (SSD) as the index of fit. The lower the SSD, the better the fit. One limitation of this modelling approach is that it uses overall performance across the entire experiment to fit the data, which ignores any changes in representation that might occur over time. In order to better characterize representation use over time, we adopted a stage modelling approach. In this approach, we each took 10 blocks of learning as a stage and calculated the degree of fit for each stage independently. This allowed us to identify changes in the use of prototype and exemplar representations across the whole learning process [30]. Model fitting was performed using a custom code programmed in MATLAB (R2009a) software. The code for each of these models is available in the Appendix A.

## 3. Results

### 3.1. Learning to Criterion

JASP 0.10.2 statistical software was used for data analysis. Mean accuracy was calculated for each participant for each block. The number of participants and the mean of learning blocks to reach the criteria were calculated. A total of 29 of the 44 participants met the learning criteria earlier than block 70. Across all participants, the mean number of blocks to reach criterion was 27.35 (*sd* = 15.61, 10 trials per block). Because there was a large amount of variability between participants, we divided participants into “good” and “poor” learners based on blocks to criterion, with those reaching criterion in 27 or fewer blocks categorized as “good learners” (*n* = 18) and those requiring more than 27 blocks as “poor” learners (*n* = 26). This allowed us to assess whether patterns in representation use were characteristic of all participants or primarily for good or poor learners alone.

### 3.2. Representations: Diagnostic Stimulus Classification Evidence

As discussed in the Introduction, responses to stimuli A1 and A2 can be used to identify the representations used by participants. A1 is a good match for Category A based on prototype similarity, but a poor match based on exemplar matching. Conversely, A2 is a good match for Category A based on exemplar similarity, but a poor match based on prototype similarity. Therefore, if the accuracy rate of A1 is higher than that of A2, it indicates that participants are using a prototype representation, whereas if A2 is higher than A1, it indicates an exemplar representation. A1 and A2 accuracy rates (calculated as the mean of Category A response) are shown in Table 2.

Overall, both A1 and A2 tended to be categorized as Category A with no significant difference in A1 and A2 accuracy across the whole experiment, *t* (43) = 0.73, *p =* 0.469, Cohen’s d = 0.11. To examine evolution of representation over time, we divided the experiment into halves. Because participants varied in the number of blocks to criterion, the two halves were determined for each participant individually. For example, for a participant who reached criterion in 20 blocks, blocks 1–10 were defined as the first half, and blocks 11–20 as the second half, whereas for a participant who reached criterion in 60 blocks, the first half included blocks 1–30 and the second half blocks 31–60. A2 (stimulus: A1 vs. A2) ×2 (half: first vs. second) repeated measures ANOVA found no significant main effect of A1–A2, *F*(1, 43) = 1.16, *p* > 0.288, *η^2^* = 0.03, but a main effect of half, *F*(1, 43) = 33.25, *p* < 0.001, *η^2^*=0.44, such that accuracy increased in the second half. The interaction was also significant, *F*(1, 43) = 4.14, *p =* 0.048, *η^2^* = 0.09, which appeared to be driven by a difference between A1 and A2 that emerged in the second half. A simple effect test revealed a significant difference between A1 and A2 in the second half, *F*(1, 43) = 4.08, *p =* 0.050, *η^2^* = 0.09, but not in the first half, *F*(1, 43) = 0.21, *p =* 0.649, *η^2^* = 0.05. This result suggests that as they learned, participants shifted to reliance on prototype representations.

We also compared good and poor learners. A2 (stimulus: A1 vs. A2) ×2 (Learners: Good vs. Poor) repeated measures ANOVA found a significant interaction, *F*(1, 42) = 6.96, *p* = 0.012, *η^2^* = 0.14. Further simple effect test results showed that the difference was mainly due to the good learners showing a significant difference between A1 and A2, *F*(1, 42) = 4.84, *p* = 0.033, *η^2^* = 0.10, while the difference was not significant for the poor learners, *F*(1, 42) = 2.19, *p* = 0.146, *η^2^* = 0.05. This result shows that good learners were especially likely to use prototype representations.

### 3.3. Representation: MPM and GCM Model Fitting Evidence

Each participant was categorized as primarily MPM or primarily GCM based on the fit of the model across all blocks that they completed (see above for model fitting details). Among them, the participants whose behavior was best fit by MPM (*N* = 18) reached the criterion in 24 blocks on average, while the participants whose behavior was best fit by GCM (*n* = 26) reached the criterion in 34 blocks on average, a statistically marginal significant difference, *t*(33) =1.89, *p* = 0.068, Cohen’s d = 0.65. The learning curves for MPM and GCM participants are shown in Figure 2; as can be seen, the differences between strategies emerges around block 11. We further examined model fits in the good and poor learner groups separately (not illustrated). A higher proportion of the MPM group were good learners (61%), and a higher proportion of the GCM group were poor learners (73%). A Chi-square test revelated that good learners were significantly more likely to be categorized as MPM users, while poor learners were significantly more likely to be categorized as GCM, χ*^2^* = 5.14, *df* = 1, *p* = 0.023.

We further examined the development of representation use over time by fitting the model to each participant across groups of 10 blocks (100 trials). As shown in Table 3 and Figure 3, the proportion of participants fit by the MPM model increased across blocks. There was also a tendency for the initial blocks to be best fit by the MPM model.

### 3.4. Dimensional Weighting

We examined the weighting of each dimension for the MPM and GCM participants across the two halves of the experiments, as shown in Table 4. Because participants varied in the number of blocks to criterion, the two halves were determined for each participant individually. For example, for a participant who reached the criterion in 20 blocks, blocks 1–10 were defined as the first half, and blocks 11–20 as the second half, whereas for a participant who reached the criterion in 60 blocks, the first half included blocks 1–30 and the second half blocks 31–60. It is helpful to compare the weightings made by the participants to the diagnostic value of each dimension for an ideal observer, which are D1 = 0.7, D2 = 0.6, D3 = 0.8 and D4 = 0.6. Thus, a participant whose behavior matches the diagnostic value of the dimensions should show the following pattern of weighting: D3 > D1 > D2 = D4.

To compare the overall fits in the MPM and GCM groups, A 2 (group: MPM or GCM) × 4 (dimension: D1-D4) ANOVA was performed. The results revealed a significant main effect of dimension, *F*(3, 129) = 14.274, *p <* 0.001, *η^2^* = 0.249; no main effect of model group, *F*(3, 129) = 0.140, *p* = 0.710, *η^2^* = 0.003; and a significant interaction, *F*(3, 129) = 13.864, *p <* 0.001, *η^2^* = 0.244. Post hoc tests showed differential weightings of the dimensions in participants within each model group. Participants in the GCM model group showed a pattern of weightings such that D3 > D1 > D4 = D2 (*p* values for each pairwise comparison between adjacent dimensions: *p =* 0.000, *p =* 0.026, *p =* 0.163). In contrast, participants in the MPM group showed the following pattern: D3 > D1 = D4 = D2 (*p =* 0.031, *p =* 0.832, *p =* 0.277). To summarize, both groups showed weighting of D3 that correctly reflects its importance. However, the MPM group overall did not make further distinctions between the dimensions, whereas the GCM group did.

In order to examine how the weighting of dimensions changed across learning in each model group, separate 2 (experiment half: first, last) × 4 (dimension: D1–D4) ANOVAs were performed. For the MPM group, the dimension main effect was significant, *F*(3, 129) = 12.86, *p <* 0.001, *η^2^* = 0.23; the experiment half main effect was not significant, *F*(3, 129) = 1.31, *p* = 0.259, *η^2^* = 0.03; and the interaction effect was significant, *F*(3, 129) = 5.67, *p* = 0.002. *η^2^* = 0.12. Post hoc simple effect tests showed that, for the first half, D3 = D1 = D4 > D2 (*p =* 0.831, *p =* 0.936, *p =* 0.000), and for the second half, D3 > D1, D4 and D2 (*p =* 0.000, *p =* 0.000, *p =* 0.000) and D1 > D2 (*p =* 0.025). Overall, this pattern of results indicates that MPM participants began to weight the more important critical dimensions over less critical dimensions early in training, and developed a strong weighting for the most critical dimension, D3, as they neared criterion performance.

For the GCM group, the dimension main effect was significant, *F*(3, 129) = 10.54, *p <* 0.001, *η^2^* = 0.20; the half main effect was not significant, *F*(3, 129) = 0.12, *p* = 0.660, *η^2^* = 0.05; and the interaction effect was significant, *F*(3, 129) = 3.60, *p* = 0.019, *η^2^* = 0.08. Post hoc simple effect tests showed that, for the first half, D3 > D1 = D4 = D2 (*p =* 0.03, *p =* 0.865, *p =* 0.055), and for the second half, D1 = D3 > D4 = D2 (*p =* 0.663, *p =* 0.024, *p =* 0.352). This pattern of results indicated a correct strongest weighting of D3 early in training, but a spreading of attention across dimensions as they approached the criterion. A comparison of both MPM and GCM groups indicates that as they approached the criterion, the MPM participants became more focused on the most important criterion, whereas GCM participants became less focused.

## 4. Discussion

Overall, we found that participants who learned the 5/5 category benefited from using a prototype representation. Participants whose performance was best fit by the MPM performed better than those whose performance was fit by GCM. Good learners were more likely to use prototype representation (as evidenced by A1–A2 stimulus classification and by computational modeling). All participants showed a shift towards prototype representation across the time course of learning. Finally, when we examined the weighting of individual dimensions, we found that participants fit by the MPM weighted the most diagnostic feature, D3, more strongly in the second half than did participants fit by GCM.

### 4.1. Representational Change across the Process of Category Learning

By employing phased model fitting analysis, we observed fluctuations in the percentage of participants best fitted by prototype models throughout the entire learning process. Initially, within the first 10 blocks, there was a tendency towards prototype representations, which subsequently became more prominent over time. However, the predominance of prototype representations was less dominant in the intermediate blocks, resulting in a tendency toward a U-shaped curve. Liu et al. argued for three stages of learning, each characterized by a different learning strategy or representation [16,31]. In Liu et al.’s theory, the first phase is the rule strategy phase, which seeks to identify the maximum diagnostic feature dimension to construct a unidimensional rule. Accuracy of the participants at this stage is usually close to the diagnostic value of the feature dimension identified and, therefore, performance is likely to be fit by a prototype model. The second stage is the rule plus exception strategy stage, in which the participants are not satisfied with the accuracy of the single-dimension rule and try to remember some examples of the exception. The classification accuracy of this stage is slightly higher than the diagnostic level, but less than 90%. The increased emphasis on the exception stimuli can result in a relative increase in the fit of the GCM model during this phase, as seen in the intermediate blocks in Figure 3. The final phase is the information integration phase, in which additional dimensions are included in the representation; this stage can result in increased fit of prototype models as the structure of the category becomes fully learned. The classification accuracy of this stage is over 90%.

### 4.2. Between Task Structure and Representations: Implications for Strategy Use in Category Learning

In this paper, we have focused on the representation of category structure learned by participants, focusing on prototype and exemplar representations. However, what accounts for the differences in participants’ representations? This requires an examination of the different strategies that can be brought to bear by participants to learn categories, of which a number have been proposed.

First, there are attentional mechanisms. Increased salience of dimensions may support ease of visual parsing of the stimulus and identification of critical features. Many theories postulate that category learning is supported by learned differences in attention to features and dimensions, and that these attentional differences determine the basis of categorization [7]. We used computational modeling to examine the weights of different dimensions and found, consistent with previous research, that prototype processing was associated with greater weighting of diagnostic features that reflect the underlying dimension weights within the category structure itself [4,5,14]. Other research has questioned the nature of this weighting and whether it truly reflects visual attention. Taking eye-tracking as a measure of visual attention, Rehder and Hoffman [13] found that eye fixations better reflected the distributed attention to features characteristic of GCM, rather than the highly focused attention on critical dimensions of MPM. Our results are consistent with greater dimensional weighting occurring when learning is best supported by prototypes, but readers should note the caveat that these weights may not be diagnostic solely of visual attention and may be influenced by later decisional processes.

In addition to attentional mechanisms, another factor that affects the type of representation may be the working memory capacity. Cognitive Load Theory holds that working memory has a limited capacity to process only 5–9 basic pieces of information or blocks of information at a time, and when processing information simultaneously, the interaction between elements stored in it also requires a working memory space [32]. Participants may attempt to apply different cognitive strategies during learning which can affect the representations acquired. A memorization strategy would be expected to result in behavior best fit by an exemplar model. In many category learning tasks, memorization has limited efficacy because these tasks include a large number of different stimuli that can overwhelm working memory capacity. Liu et al. [31] showed that if participants were told the number of unique stimuli in advance, they tended to acquire exemplar representations. If it exceeds its working memory capacity, they may tend to acquire prototype representations. Even for categorization tasks in which performance cannot be fully accounted for by memorization, memorization can serve as one component in learning. For example, “rule plus exception” theories propose that exception stimuli are memorized [33]. Future research should further examine the effects of working memory capacity on learning in the 5/5 task.

A final mechanism is reinforcement learning. Reinforcement learning has been shown to account well for complex multivalued and continuous valued dimensions, such as are typical of information integration category learning tasks. According to the COVIS model [34], reinforcement learning serves to map small regions of perceptual space to categories. The prediction of what apparent category representation will result will depend on the interaction of attention with perceptual space. If attention is directed to some dimensions more than others, then the overall patterns of categorization behavior may mimic prototype learning. To the extent that perceptual space includes equal attention to all features, one would predict exemplar representations [35].

## 5. Conclusions

We developed a novel 5/5 category learning task that was modified from the 5/4 task to reduce ambiguity and to increase category coherence. Our primary goal was to examine how use of prototype and exemplar representations changed over the time course of learning, which has rarely been studied in previous research. We found that use of prototype representations increased over training. We also found that use of prototype representations was associated with better learning overall than use of exemplar representations.

## Figures and Tables

**Figure 1 behavsci-14-00470-f001:**
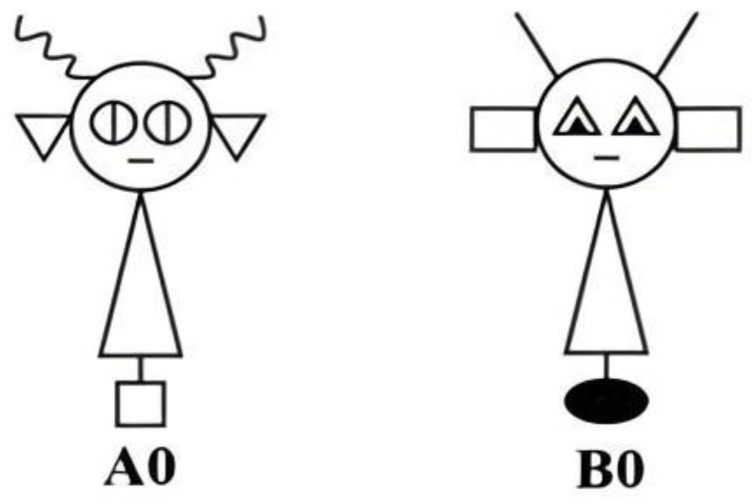
Sample prototype stimuli from the Experiment.

**Figure 2 behavsci-14-00470-f002:**
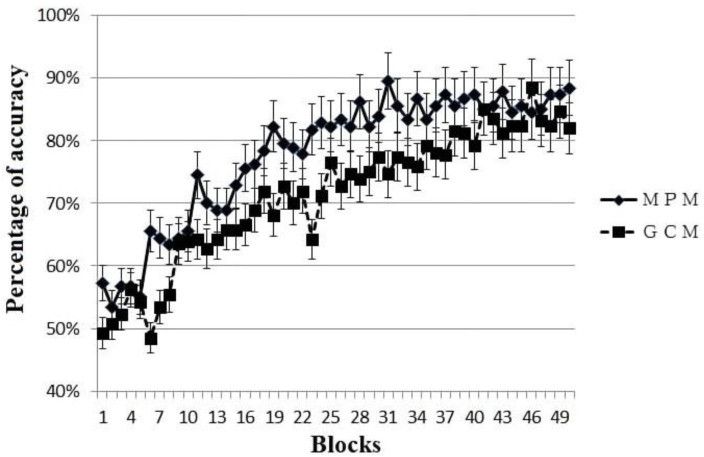
Learning curves for participants with behavior best fit by MPM (multiplicative prototype model) or GCM (generalized context model) as calculated across the entire time course of the experiment. Data are plotted only for the first 50 blocks due to the limited number of participants who did not reach criterion until after block 50.

**Figure 3 behavsci-14-00470-f003:**
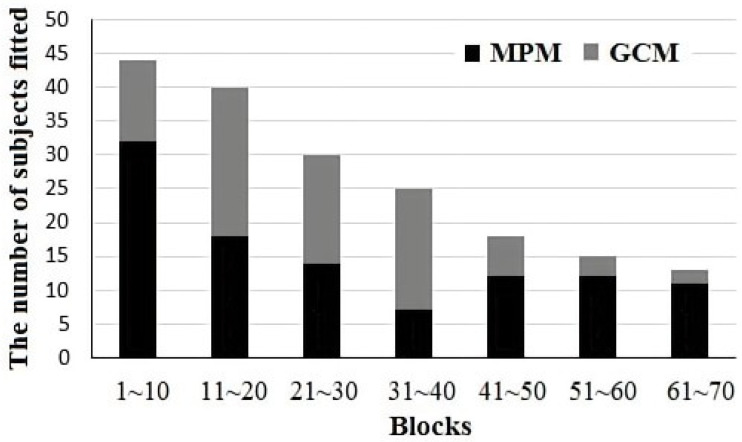
Histogram of number of participants in each group of 10 blocks (100 trials) best fit by the MPM and GCM strategies. Note that the number of participants included decreases over blocks as participants reach the criterion.

**Table 1 behavsci-14-00470-t001:** “5/4” and “5/5” category structure.

	A					B			
	D1	D2	D3	D4		D1	D2	D3	D4
A1	1	1	1	0	B1	1	1	0	0
A2	1	0	1	0	B2	0	1	1	0
A3	1	0	1	1	B3	0	0	0	1
A4	1	1	0	1	B4	0	0	0	0
A5	0	1	1	1	B5	1	0	0	1
A0	1	1	1	1	B0	0	0	0	0

Note: Feature definitions for the 5/4 and 5/5 structures. The 5/4 structure differed from the 5/5 structure in that stimulus B5, indicated by the surrounding box, is not included in the 5/4 structure. A0 and B0 are prototypes of Category A and B, respectively. High similarity exemplar pairs (3 feature overlap) are stimuli A1(1110) and B1(1100), A1(1110) and B2(0110), A2(1010) and A1(1110), and A2(1010) and A3(1011).

**Table 2 behavsci-14-00470-t002:** Categorization of diagnostic stimuli.

	Whole	First Half	Second Half	Good Learner (*N* = 18)	Poor Learner (*N* = 26)
A1	0.71(0.17)	0.64(0.20)	0.85(0.18)	0.77(0.16)	0.67(0.17)
A2	0.70(0.15)	0.66(0.18)	0.77(0.24)	0.72(0.16)	0.68(0.14)

Note: Proportion of classification of stimuli A1 and A2 as members of Category A. A1 stimuli are diagnostic of a prototype representation, A2 of an exemplar representation. Standard deviations are indicated in parentheses.

**Table 3 behavsci-14-00470-t003:** Stage fitness of MPM and GCM across blocks.

	Blocks	1~10	11~20	21~30	31~40	41~50	51~60	61~70
	N	44	40	30	25	18	15	13
MPM	SSD	0.217	0.256	0.253	0.329	0.222	0.274	0.133
	Fitness	0.73	0.45	0.47	0.28	0.66	0.80	0.85
GCM	SSD	0.237	0.241	0.240	0.299	0.282	0.323	0.234
	Fitness	0.27	0.55	0.53	0.72	0.33	0.20	0.15

Note: Table indicates, for each grouping of 10 blocks (100 trials each), the proportion of participants best fit by each model and the mean degree of fit measured by SSD. For this analysis, participants were categorized as MPM or GCM on each set of blocks separately, rather than categorized based on their overall fit across the whole experiment. SSD: sum of squared deviations, the measure of model fit. Fitness: proportion of participants best fit by the model in the particular group of 10 blocks.

**Table 4 behavsci-14-00470-t004:** Fitness and dimension weighting for MPM and GCM participants.

Model		Fitness	D1	D2	D3	D4	*C*
MPM	1st half	0.535	0.311	0.055	0.331	0.303	0.230
	2nd half	0.973	0.201	0.079	0.532	0.188	0.395
	Total	0.103	0.233	0.103	0.521	0.144	1.242
GCM	1st half	1.430	0.223	0.088	0.477	0.212	0.477
	2nd half	0.978	0.394	0.098	0.351	0.158	2.228
	Total	0.106	0.226	0.158	0.403	0.214	4.125

Note: Halves were defined for each participant individually based on their blocks to criterion. *N* = 44. D1–D4: Dimension 1–Dimension 4 weights, respectively (dimensional decision weight (attention weight), W). C: sensitivity parameter.

## Data Availability

Our research data will be made available upon reasonable request.

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
