# Peer review of "Prototype or Exemplar Representations in the 5/5 Category Learning Task"

_behavsci, 2024, doi:10.3390/bs14060470_

Round 1

Reviewer 1 Report

Comments and Suggestions for Authors

Chen et al. introduce a novel “5/5” categorization task and characterize learning in the task with prototype- and exemplar-based computational models. The results suggest that participants who more successfully learned in the task relied more heavily on prototype-based representations and that learning became more prototype-based across blocks.

I enjoyed reading the paper and I applaud the use of modeling. Overall, I think the paper holds promise, but I do have a number of concerns that should be addressed prior to publication, as I outline below.

My greatest concern is that the larger implications of the study are not clear to me. The authors present a new categorization task variant in the 5/5 task (which includes a new stimulus type compared to the 5/4 task), but they don’t compare it to the 5/4 task or any other task, so it’s not super clear whether the task is helping us better understand categorization. I think it would improve the paper to include a condition with the 5/4 structure, particularly as the authors claim that the 5/5 task is an improvement in that it “eliminates the ambiguity about the prototype of category B… and increases the diagnostic value of dimension 2.” It would be great if we could assess those claims directly by applying the same analyses that the authors already have to performance on the 5/4 task. If that’s not possible, perhaps the authors could do a better job of comparing their findings to what is known about the 5/4 task from previous studies.

The authors compare exemplar-based and prototype-based models of task performance, which I applaud, but again, it’s not clear to me what the modeling tells us about categorization in general, other than what model fits this particular task better. The problem struck me immediately in the abstract, the last sentence of which is “We found the schematic robot-like stimuli was characterized by use of prototypes.” To me, that’s a very narrow viewpoint -- it doesn’t mention the possibility of any larger implications or give an indication that implications will even be discussed. The authors state on page 2 that “Our goal was to test how task characteristics affect whether participants rely on prototype or exemplar representations,” and the authors did find evidence of greater reliance on prototypical representations, at least for participants who were more successful. They also modeled performance in specific blocks to investigate representational change over the course of learning, which I thought was very neat. I couldn’t tell, however, whether this was a novel analysis – if so, perhaps the authors could emphasize that more. In the paper’s conclusion, the authors say that the study “highlights the importance of multidimensional stimulus materials in category learning and indicate that these factors should be incorporated into future experimental work and theoretical development.” However, it seems to me that researchers have been using multidimensional stimulus materials in this kind of research for decades, including in the 5/4 task, so, again, the authors just need to do a better job of clarifying the novelty and importance of this work.

More specific comments/concerns:

Introduction

·         Lines 124-129 say that the authors “hypothesized that memory capacity would affect the likelihood of ultimately developing exemplar or prototype representations. More specifically, we propose that within working memory capacity, stimuli would be learned via exemplar representations, but that as stimuli became more complex and more dimensions, the prototype representations would become more dominant.” This led me to believe that working memory capacity would be measured or manipulated in some way, but it was not. As a result, I would either remove this section or substantially reword, especially because I don’t see how the study addressed stimulus complexity, either. Just to be clear, I do think working memory capacity is important to discuss, but I don’t see how this study addresses its impact directly, so I wouldn’t include it in the hypotheses.

·         The “Current Study” section (lines 140-143) indicates that the authors hypothesized that the 5/5 category structure would result in prototype representations, but it’s unclear why they made that hypothesis. Is it something about the 5/5 task in particular (as opposed to 5/4 or other task structures)?

Methods

·         Lines 157-160: “At the same time, due to the working memory capacity, the stimuli are multi-dimensional and cannot be completed by memory alone. We predicted that it would facilitate selective attention to dimensions and therefore we would see use of prototype representations.” Two questions/comments about that: How do the authors know that the task cannot be completed by memory alone (are they calculating the number of dimensions/features that must be encoded in working memory and comparing that to 7 plus/minus 2)? Also, it seemed odd to me to make the prediction about selective attention due to prototype representations, as selective attention is not at odds with exemplar theory.

·         Procedure: To what extent is the procedure based on previous work? Is this the standard procedure for the 5/4 task?

·         Line 181: Do the authors mean that 6x8cm is the size of the stimuli instead of the size of the screen?

·         Line 215: the authors say that similarities are raised to a gamma parameter, but that is not included in the equation. I’m assuming that’s because they set the value of gamma to 1 so it would have no effect, but it should be more clear.

·         Speaking of parameters, I’d like to see a summary of each of the free parameters for both models.

Results

·         According to my calculations, 15 participants never met the learning criteria, even after 70 blocks, but those participants are still included in the analyses. My concern is that some or all of those participants may have been making random responses either because they didn’t learn at all, or did not understand the task, or weren’t motivated, etc., any of which would make the data uninterpretable. Is there some way that we can confident that these participants were even trying and doing the task at all? Maybe you could compare each participant’s accuracy in the last few blocks to chance (50%) with a binomial test, or something like that, to determine if they should even be included in the analyses at all.

·         Does the pattern of results change if you use a median split for good versus bad learners instead of a mean split (lines 256-262)?

·         I thought that Table 3 and Figure 3 showed an interesting result that wasn’t really addressed by the authors, which is that there seems to be a U-shaped curve, where MPM fit more participants early in the task (blocks 1-10), and later in the task (blocks 40+), whereas more participants were better fit by GCM in the middle bocks. Any ideas as to why that might be? (The authors suggest that the proportion of participants better fit by MPM simply increased across blocks, which ignores the high proportion in blocks 1-10.)

Discussion

·         What is it about the task that encourages prototype-based learning? Is it that the 5/5 task increases the coherence of the categories (lines 100-103 discuss the Bowman and Zeithamova study along these lines)? Does this task encourage more prototype-based learning than 5/4?

Conclusion

·         Last sentence: “The result highlights the importance of multidimensional stimulus materials in category learning and indicate that these factors should be incorporated into future experimental work and theoretical development.” I’m confused by this because it seems to me that probably the majority of category learning studies use multidimensional stimulus materials, including the 5/4 task and many others, so I’m no sure why that needs to be highlighted. I’m also confused as to what the authors mean by “these factors.”

·         Overall I was underwhelmed by the conclusions – I think the authors could/should make their novel contributions more clear.

Author Response

Dear reviewer, thank you for your time and your comments, We have modified the relevant content according to your suggestion, please refer to the attachment for details. Wish you all the best.

Reviewer 2 Report

Comments and Suggestions for Authors

Categorization is a fundamental learning ability that enables individuals to rely on prior experience when encountering new environmental stimuli. The question of whether this process is based on prototype or exemplary mechanisms is still debatable. The manuscript titled "Prototype or Example Representations in the 5/5 Category Learning Task" seeks to address this question by presenting a study that examines a new 5/5 categorization strategy, which aims to resolve the ambiguity associated with the previous 5/4 categorization method. The authors applied two distinct computational models to assess whether individuals rely on exemplar or prototype representation for the efficient categorization of various visual tasks. Their results indicate that prototype representation was the preferred approach when utilizing the 5/5 category.

In general, this paper is well-written and clearly structured, providing a detailed methodology and sufficient references. The study's revelation that prototype representation, rather than exemplar representation, is favored for efficient categorization, which advances our understanding of the underlying mechanisms in categorization. I have no further questions or suggestions prior to its publication.

Comments on the Quality of English Language

No obvious grammar error was found. 

Author Response

Dear reviewer, thank you for your time and your comments, we will continue to work hard and wish you all the best.

Reviewer 3 Report

Comments and Suggestions for Authors

Dear Authors, 

I enjoyed reading your manuscript. The manuscript is highly informative and well-organized- Every aspect of scientific research is considered accordingly, the literature matches the research idea and the discussions. The results can contribute to the current literature. Hope to read more from you. 

Good luck 

Author Response

Dear reviewer, thank you for your efforts in reviewing this paper and affirming the author, we will make a better paper with greater enthusiasm. Wish you all the best.

Round 2

Reviewer 1 Report

Comments and Suggestions for Authors

The authors addressed my concerns and I am pleased with the new version of the manuscript. Well done!